# Estrogen receptor positive breast cancers have patient specific hormone sensitivities and rely on progesterone receptor

Valentina Scabia[1], Ayyakkannu Ayyanan [1], Fabio De Martino[1], Andrea Agnoletto[1], Laura Battista[1], Csaba Laszlo[1,7], Assia Treboux[2], Khalil Zaman[2], Athina Stravodimou[2], Didier Jallut[3], Maryse Fiche[4], Philip Bucher [1], Giovanna Ambrosini[1,5], George Sflomos [1] & Cathrin Brisken [1,6 ✉]

Estrogen and progesterone receptor (ER, PR) signaling control breast development and impinge on breast carcinogenesis. ER is an established driver of ER + disease but the role of the PR, itself an ER target gene, is debated. We assess the issue in clinically relevant settings by a genetic approach and inject ER + breast cancer cell lines and patient-derived tumor cells to the milk ducts of immunocompromised mice. Such ER + xenografts were exposed to physiologically relevant levels of 17-β-estradiol (E2) and progesterone (P4). We find that independently both premenopausal E2 and P4 levels increase tumor growth and combined treatment enhances metastatic spread. The proliferative responses are patient-specific with MYC and androgen receptor (AR) signatures determining P4 response. PR is required for tumor growth in patient samples and sufficient to drive tumor growth and metastasis in ER signaling ablated tumor cells. Our findings suggest that endocrine therapy may need to be personalized, and that abrogating PR expression can be a therapeutic option.

[1] Swiss Institute for Experimental Cancer Research, School of Life Sciences, Ecole Polytechnique Fédérale de Lausanne, CH-1015 Lausanne, Switzerland. [2] Breast center, Lausanne University Hospital (CHUV), CH-1011 Lausanne, Switzerland. [3] Réseau Lausannois du Sein, Rue de la Vigie 5, CH-1004 Lausanne, Switzerland. [4] International Cancer Prevention Institute, CH-1066 Epalinges, Switzerland. [5] Bioinformatics Competence Center (BICC), UNIL/EPFL, CH-1015 Lausanne, Switzerland. [6] Breast Cancer Now Research Centre, Institute of Cancer Research, London, UK. [7] Present address: Philip Morris International R&D, Philip Morris Products S.A., Neuchâtel, Switzerland. ✉email: cathrin.brisken@epfl.ch

Breast cancer (BC) is the most frequently diagnosed cancer worldwide[1]. More than 70% of all BCs are classified as ER + based on the detection of ER expression by immunohistochemistry in at least 1% of the tumor cells. Reproductive factors such as early menarche, late menopause, and late pregnancies are known to affect breast cancer risk through changing exposures to the ovarian hormones, 17-β-estradiol (E2), and progesterone (P4)[2]. Exposures to exogenous hormone receptor agonists in the context of hormonal contraception and hormone replacement therapy also impact risk[3,4] and have linked PR signaling to BC risk[5,6] and tumor progression[7].

ER signaling is a key driver of ER + breast carcinogenesis and inhibition of ER signaling is the mainstay of ER + BC therapy and has substantially improved patient survival[8]. The use of PR antagonists for patients with advanced BC has been unsuccessful, largely due to severe side effects linked to the low specificity of the antagonists[9]. While PR blockade was shown to have cytostatic effects and, in some cases, led to tumor regression in combination with tamoxifen[10–12], contrasting evidence has also been provided that PR activation blocks estrogen-induced tumor growth[13,14].

Studies with BC cell line models have revealed important crosstalk between ER and PR signaling, both at the genomic and protein level[15,16]. Moreover, ligand-activated PR modulates ER action by redirecting its genomic targeting resulting in decreased or increased proliferation, increased EMT features, and changes in the rate of translation[12,14,17–21]. On the other hand, unliganded PR can govern *ESR1* expression by regulating DNA methylation[22]. As *PGR* is an ER target gene[23], endocrine therapies targeting ER result in loss of PR expression, precluding the analysis of the role of either receptor independently.

A further obstacle to our understanding of the complex roles of ER and PR signaling in BC has been the lack of adequate models. Most current knowledge about ER and PR signaling stems from in vitro experiments, in which hormone-sensitive cell lines are first hormone-deprived and subsequently stimulated with either one or both hormones. In vivo, however, hormone levels never equal zero but fluctuate continuously at different levels. Although ER + tumors represent the vast majority of BCs, there is a limited number of ER + BC cell lines, most of which were established from pleural effusions, i.e., very advanced disease[24]. Out of these ER + cell lines very few, like MCF7 and T47D cells can be established as subcutaneous or mammary fat pad tumor xenografts, and this only under the condition that the host mouse is supplemented with exogenous E2. This results in premenopausal or even higher E2 levels in the animals when most ER + BCs arise in postmenopausal women, who have barely detectable E2 levels. As ER signaling is exquisitely dose-dependent, this creates experimental conditions of reduced clinical relevance.

Patient-derived xenografts (PDXs) are increasingly used in cancer research because they model more closely intra- and interpatient heterogeneity than the widely used cell line models. However, they were notoriously difficult to establish from ER + BCs with only 2.5% efficiency and a bias for aggressive phenotypes[25]. We and others have recently shown the grafting of hormone-sensitive BC cells (both established cell lines and patient-derived) into the milk ducts of adult NOD.Cg-Prkdc*scid* Il2rg*tm1Wjl*/SzJ (*NSG*) female mice enable high take rates and in vivo growth without exogenous hormone supplementation[26–30]. In these xenograft models, micrometastases develop in clinically-relevant organs, such as the brain, lungs, liver, and bone[30,31], and biological features of the patient tumors[27] and of specific subtypes[29] are retained.

Here, we show patient-specific responses of ER + BC cells to physiologically relevant levels of E2 and progesterone (P4) and dissect the role of PR genetically in vivo.

## Results

**Physiologic hormone exposures promote ER + BC xenograft growth.** To determine the effects of physiologic E2 and P4 exposures as they occur in premenopausal women during menstrual cycles on ER + breast carcinogenesis in vivo, we engrafted immunocompromised NOD.Cg-Prkdc*scid* Il2rg*tm1Wjl*/SzJ (*NSG*) females by direct injections into the milk ducts with widely used ER + BC cell lines and fresh patient tumor-derived cells, genetically labeled with a dual luciferase-RFP reporter (Fig. 1a). Successful engraftment and subsequent growth were ascertained by in vivo bioluminescence measurements. Several weeks later, when the average luminescence per gland was >10E6, the mice were randomized and implanted with slow-release pellets containing different doses of E2 and P4 (Fig. 1b, c). Dosage of E2 plasma levels in control pellet bearing *NSG* females showed values comparing well to those observed in postmenopausal women (7 ± 1.15 pg/ml) (Fig. 1b, shading)[32], while P4 levels varied from 0.1 to 20 ng/ml related to estrous cycle averaging 4 ng/ml when postmenopausal levels are <3.2 ng/ml[33] (Fig. 1c, shading). About 0.3 mg E2 pellets yielded plasma E2 and P4 levels approximating those in premenopausal women in the follicular phase (Fig. 1b, d)[32] and combined E2 0.3 mg and P4 20 mg pellets yielded luteal phase E2 and P4 levels (Fig. 1c, shading). All treated mice remained in good health; the bodyweight of control and E2-treated mice increased by 8% over the 45 days, whereas P4 and E2 + P4 treated mice showed ~20% weight gain (Fig. 1e). E2 increased MCF7 xenograft growth (Fig. 1f) and metastatic load (Fig. 1g) in a dose-dependent fashion across different organs (Fig. 1h, i), validating that our in vivo approach is quantitative.

Next, we stimulated different cell lines derived from ER + BCs, MCF7, and T47D, in which the *PGR* locus is amplified[34], HCC1428, and the lobular MDA-MB-134-VI cells with E2, P4, and E2 + P4. The effects on in vivo growth were analyzed during treatment and at the endpoint. The relative increase in tumor growth in response to the different hormone treatments was cell line-specific (Fig. 1j); MCF7 cells were exquisitely E2-sensitive and had the highest growth rates while the lobular MDA-MB-134-VI cells only showed a proliferative response to P4.

**Hormone exposures promote ER + BC growth in a patient-specific fashion.** To address whether the heterogeneous proliferative response of the ER + cell lines reflects the properties of ER + patient tumors, we generated xenografts with cells from nine patients with either untreated primary tumors or pleural effusions (Fig. 2 and Table 1). Freshly isolated patient-derived cells were lentivirally transduced with *RFP:Luc2*. The resulting mixtures of infected and uninfected cells were injected without selection via the teats and serially propagated for one or two generations in vivo over 20–36 months. When a sufficient amount of cells had grown, experimental mice bearing a PDX in two or three of their mammary glands were randomized for 60-day-hormone treatments. Growth was analyzed by comparing the slopes of in vivo bioluminescence and the relative increase at endpoint. Two PDXs (T99 and T113), were exclusively stimulated by P4. One PDX, T110, responded exclusively to E2, the remaining six all responded to E2 stimulation but sometimes more or less than to P4 or to the combined treatment (Fig. 2). At variance with the others, the PDX T109 was actually inhibited by the combined treatment (Fig. 2). Thus, each cell line and each individual PDX had distinct proliferative responses to E2 versus P4 and E2 + P4. Overall, the extent of the proliferative responses elicited by any of the hormone treatments was lower in the PDXs than in the cell line-derived xenograft models.

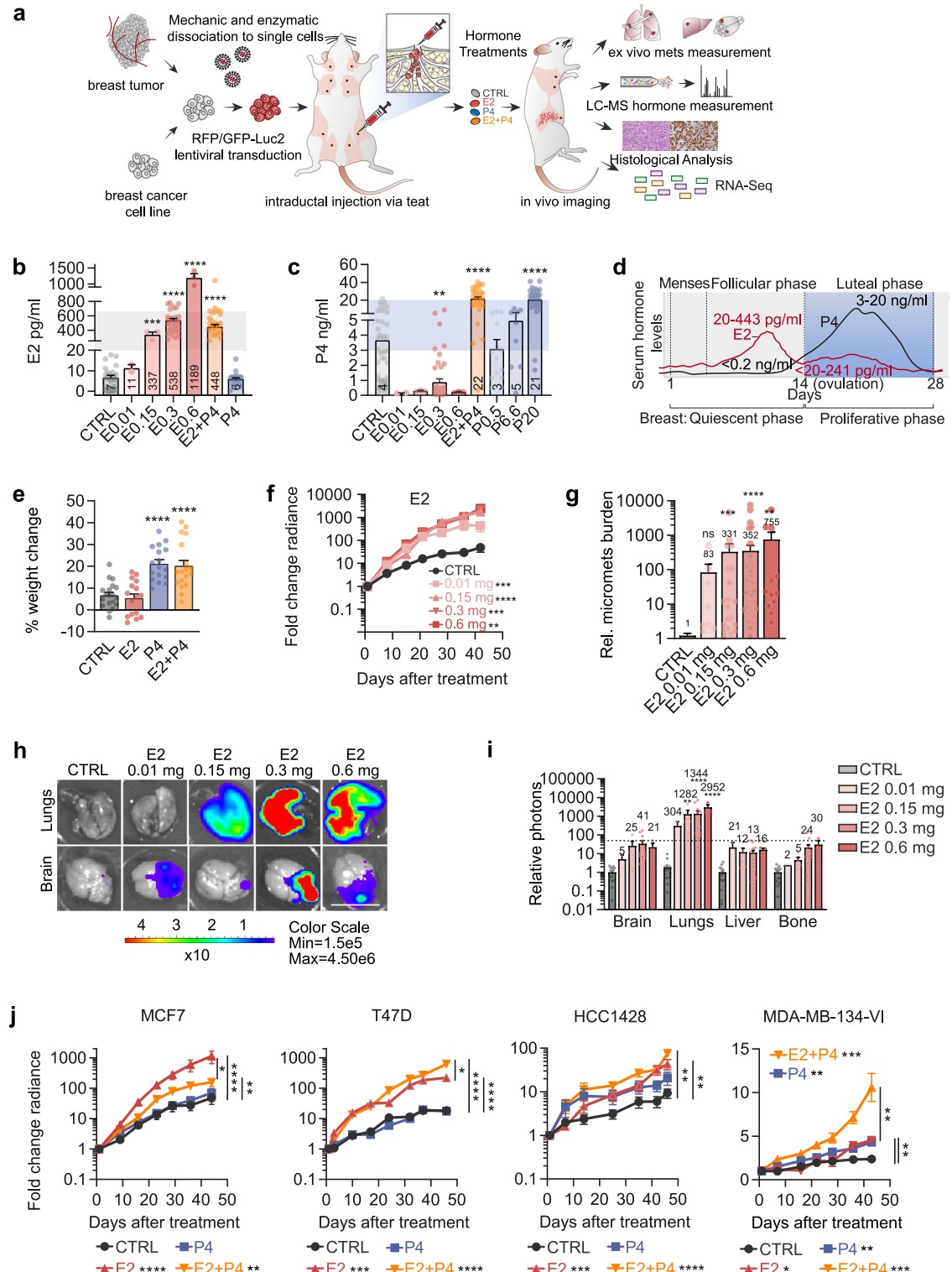

**Hormone exposure increases metastatic spread.** While early BC has an excellent prognosis, the disease becomes difficult to control when metastasis has occurred. To assess the effect of hormone exposures on the clinically important metastatic spread, ex vivo luminescence was measured on distant organs of xenografted mice at sacrifice. E2 treatment significantly increased metastasis of MCF7 cells to the lungs, as did the combined treatment (Fig. 3a). P4 potentiated the enhancing effect of E2 on brain metastasis in T47D xenografts (Fig. 3b) and showed a trend to increase seeding of HCC1428 xenografts to the bones (Fig. 3c). Overall, the metastatic load was increased by E2 and the combined E2 and P4 treatment in MCF7 and T47D xenografts (Fig. 3d–f). Individual early passage PDXs showed different trends with regard to specific hormone-induced metastatic spread (Fig. 3g). Considering all PDXs combined, E2 resulted in a fourfold increase in brain metastatic load (Fig. 3h). Overall, the metastatic load was increased 2.5-fold by E2 and twofold by P4 (Fig. 3i). The combined E2 + P4 treatment increased metastatic load 2.2-fold (Fig. 3i) when it did not significantly affect tumor growth (Fig. 3j) indicating a

**Fig. 1 Effects of physiologic E2 and P4 levels on ER + BC growth. a** Experimental scheme. Cell line- and patient tumor-derived cells are infected with lentivirus coding for luciferase-GFP or RFP and injected into the milk duct tree via the teats of *NSG* females (two to four glands per mouse). Tumor growth is measured weekly by bioluminescence. Hormone pellets are implanted subcutaneously 1–16 weeks after injection. Forty-five days (for cell lines) or 60 days (for PDXs) after pellet implantation, host mice are sacrificed, mammary glands are harvested, metastases are analyzed by ex vivo bioluminescence, blood is collected, and hormone levels are determined by LC-MS. **b** Bar plot showing E2 plasma levels at endpoint in hormonally treated mice; the shade shows E2 levels in premenopausal women. Data represent mean ± SEM. One-way ANOVA, mice n = 50, 2, 3, 41, 3, 39, and 51. **c** Bar plot showing P4 plasma levels in treated mice at an endpoint, shade shows P4 levels during the luteal phase. Data represent mean ± SEM. One-way ANOVA, mice n = 63, 2, 3, 53, 3, 42, and 28. **d** Reference values of E2 and P4 levels during women's menstrual cycle[65]. **e** Bar graph displays percent bodyweight gain during the 60-days of hormone treatment. Data represent mean ± SEM. One-way ANOVA, n = 20, 17, 17, and 17 mice. **f** MCF7 tumor growth as assessed by bioluminescence, normalized to the measurement before the treatment with increasing doses of E2 (average ± SEM). Two-way ANOVA, followed by Dunnett's multiple comparison test, tumors n = 43, 8, 11, 10, and 26. **g** Bar plot showing the relative metastatic burden of mice engrafted with MCF7 cells under different E2 treatments. Ex vivo bioluminescent signal was normalized to the average signal of each organ in control mice, average ± SEM, dots represent individual organs. Kruskal–Wallis test followed by Dunn's multiple comparison test, median glands n = 48, average n = 45, n of mice: CTRL n = 15, E2 0.01 mg n = 2, 0.15 mg n = 5, E2 0.3 mg n = 15, E2 0.6 mg n = 3. **h** Ex vivo luminescence images of brains and lungs from MCF7 engrafted mice treated with different concentrations of E2. Scale bar, 1 cm. **i** Relative ex vivo radiance in different organs of the same mice. Data represent mean ± SEM. Two-way ANOVA, followed by Dunnett's multiple comparison test. Mice, CTRL n = 15, E2 0.01 mg n = 2, E2 0.15 mg n = 5, E2 0.3 mg n = 15, E2 0.6 mg n = 3. **j** Radiance-based tumor growth curves for different ER + cell line xenografts upon treatment with E2 0.3 mg, P4 20 mg or the combination of the two relative to controls. MCF7 tumors n = 31, 21, 49, and 29; T47D tumors n = 21, 14, 29, and 24. HCC1428 tumors n = 7, 8, 7, and 6; MDA-MB-134-VI tumors n = 8, 11, 8, and 6. Control (CTRL), Estradiol (E2), Progesterone (P4).

disconnect between primary tumor growth and metastatic spread in response to hormone stimulation.

**Transcriptional responses to E2 and P4.** To gain insights into the transcriptional responses to hormone stimulation, we sequenced RNA from six early passage PDXs treated with E2, P4, or CTRL (analysis of E2 + P4 condition was precluded by too little material) with three or four biological replicates for each condition of each PDX. Unsupervised clustering separated individual patient samples based on the 500 most variably expressed genes. Within each tumor, samples clustered largely according to hormone treatment (Fig. 4a). Across the six PDXs we identified 4097 E2-regulated genes (with adjusted *p* value cutoff of 0.05, out of which 1113 have logFC <−0.5 or logFC >0.5) (Fig. 4b, c). whereas only 489 genes were differentially expressed in response to P4 (Fig. 4b, d). Gene set enrichment analysis (GSEA) for hallmark pathways showed E2 and interferon α responses, as well as apoptosis, were positively enriched and androgen response was negatively enriched following E2 exposure (Fig. 4e). P4 decreased the E2 response signature and resulted in positive enrichment of MYC targets and oxidative phosphorylation as well as signatures related to de-differentiation (Fig. 4f).

**Low MYC and AR activity determine P4-induced tumor growth.** To identify mechanisms underlying the divergent response to P4, we separated the tumors into those that proliferated in response to P4 (T111, T113, and PL-015), called responders, from those that were non (or partial)-responders (T105, T110, and T109). The extent of P4-induced global gene expression changes was larger in responders (Fig. 5a) than in non-responders (Fig. 5b). GSEA for hallmark pathways showed that both groups were positively enriched for MYC targets and androgen response (Fig. 5c, d). Epithelial to mesenchymal transition (EMT) and INFα response were positively enriched by P4 stimulation in the non-responders while negatively enriched in the responders (Fig. 5c, d). While the EMT signature points to a tumor cell-intrinsic cell cycle inhibitory mechanism, the INFα response speaks to paracrine interactions and suggests that different cell types of the microenvironment, in particular macrophages, are a major source of this cytokine, also have a role in suppressing tumor cell proliferation. To identify the factors that determine the response to P4, we analyzed the genes differentially expressed between responders and non-responders at baseline.

This comparison showed that responders had, paradoxically, lower MYC targets, androgen response, and TNFα signaling as well as increased INFα response (Fig. 5e). Thus, low MYC targets and androgen response[35], two transcriptional signatures, which themselves may reflect low PR signaling activity, are associated with an increased transcriptional and proliferative response upon P4 stimulation suggesting that increased, PR signaling activity renders cells insensitive to the ligand. The differential enrichment of the immunomodulatory pathways, TNFα and INFα signaling suggest close crosstalk between the pregnancy hormone and the innate immune response as reported recently[36,37].

**PR function in ER + BC cells in vivo.** To disentangle the contributions of ER and PR to tumor growth and metastatic spread, we recurred to using the particularly E2-sensitive and readily manipulatable MCF7 cells and lentivirally transduced them with shCTRL or shRNAs targeting either *ESR1* or *PGR*. Following drug selection, some cells were seeded for in vitro colony formation, and others xenografted intraductally. Where possible, immunoblotting was performed and ER and PR protein levels decreased below the detection limit with the knockdown (Fig. 6a). As expected, MCF7:*shESR1* cells failed to form colonies (Fig. 6b, c). However, even MCF7:sh*PGR* cells showed a 75% decrease in colony formation efficiency (Fig. 6b, c). On day 1 after intraductal injection, radiance emanating from engrafted glands was comparable for all three infectants (Fig. 6d). By day 90, the sh*ESR1* and sh*PGR* expressing MCF7 cells grew respectively 75 and 60% less than the control cells (Fig. 4e), although MCF7 cells were not growth-stimulated by P4 (Fig. 1e). The findings extended to in vivo proliferation of T47D cells (Fig. 6e). In HCC1428, sh*ESR1* had the same effect, while sh*PGR* affected growth only marginally (Fig. 6e). In four different PDXs, expression of a bi-cistronic vector sh*ESR1:luc-GFP* decreased cell proliferation by 90% and sh*PGR:luc-GFP* had a similar effect, except for one PDX (T110) (Fig. 6f). Thus, downmodulation of PR expression independently of ER expression abrogates tumor growth in a majority of ER + BC xenografts indicating that PR is widely required in ER + BC cells for tumor growth. The patient-derived BC cells are more sensitive to receptor downmodulation than the cell lines are.

**Role of PR in endocrine resistance.** *PGR* is an ER target, and ER signaling inhibition results in decreased PR expression. To assess whether sustained PR expression may contribute to endocrine

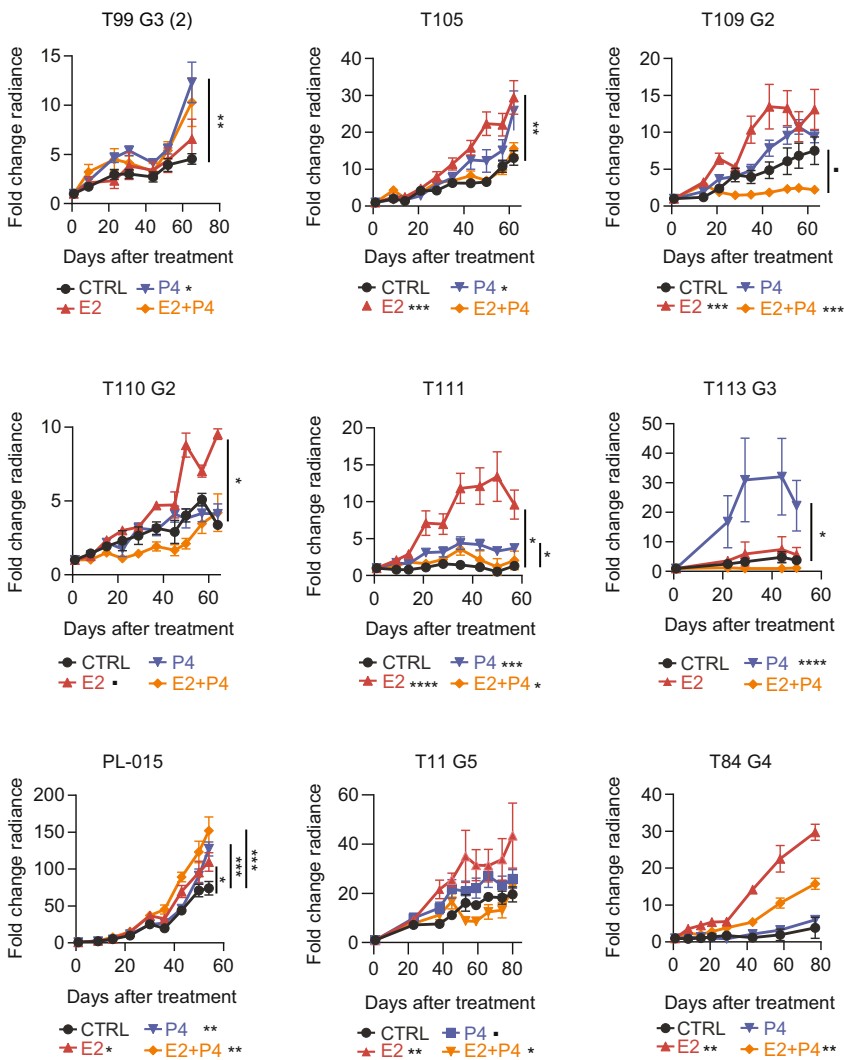

**Fig. 2 Hormone exposures and in vivo growth of ER + patient tumors.** Line charts showing in vivo growth of the different patient-derived tumor cells expressing luciferase:GFP as measured by radiance after implantation of either control or different hormone-containing pellets. Points show means of fold change of radiance in individual glands ±SEM. Statistical significance was tested by fitting a linear (or smoothing-spline when necessary) mixed effects model to the log10 transformed data. All comparisons are relative to the control group (CTRL). Legends with treatment code and asterisks are below the graphs; Median number of glands per sample T84: CTRL $n = 2$, E2 $n = 2$, E2 + P4 $n = 4$, P4 $n = 2$. T99 E2 $n = 2$, E2 + P4 $n = 4$, P4 $n = 2$. T105 E2 $n = 2$, E2 + P4 $n = 4$, P4 $n = 2$. T109 E2 $n = 2$, E2 + P4 $n = 4$, P4 $n = 2$. T110 E2 $n = 2$, E2 + P4 $n = 4$, P4 $n = 2$. T111 E2 $n = 2$, E2 + P4 $n = 4$, P4 $n = 2$. T113 E2 $n = 2$, E2 + P4 $n = 4$, P4 $n = 2$. T11 E2 $n = 2$, E2 + P4 $n = 4$, P4 $n = 2$. PL-015 E2 $n = 2$, E2 + P4 $n = 4$, P4 $n = 2$. For two-sample comparisons at endpoint, Wilcoxon rank-sum test was used on log-transformed fold change values, followed by $p$ value adjustment for multiple comparisons. For statistically significant comparisons, asterisks are on the right side of graphs. Color codes for treatment: black: CTRL, red: E2, blue: P4, orange: E2 + P4.

**Table 1 Characteristics of patient tumors.**

| Sample | Age of patient (years) | ER% | PR% | Her2 | Ki67% | Sample type | Histological tumor type |
|--------|------------------------|-----|-----|------|-------|-------------|-------------------------|
| T84 | 57 | 80 | 80 | Negative | 10 | Primary, untreated | Mixed lobular/NST |
| T99 | 57 | 100 | 70 | Negative | 20 | Primary, untreated | NST |
| T105 | 59 | 80 | 100 | Negative | 20 | Primary, untreated | Lobular |
| T109 | 44 | 100 | 100 | | 10 | Primary, untreated | NST |
| T110 | 51 | 80 | 10 | Negative | 25 | Primary, untreated | NST |
| T111 | 44 | 100 | 100 | | 10 | Primary, untreated | Mixed NST/lobular |
| T113 | 70 | NA | NA | NA | NA | Primary, untreated | NA |
| T11 | 50 | NA | NA | NA | NA | Primary, untreated | NST |
| PL-015 | 59 | 90 | 100 | Negative | NA | Ascites, treated: chemo, aromatase inhibitors, fulvestrant | NST |

The table displays characteristics of patient tumors used in this study; the last column shows the response to P4 as MIND PDX.

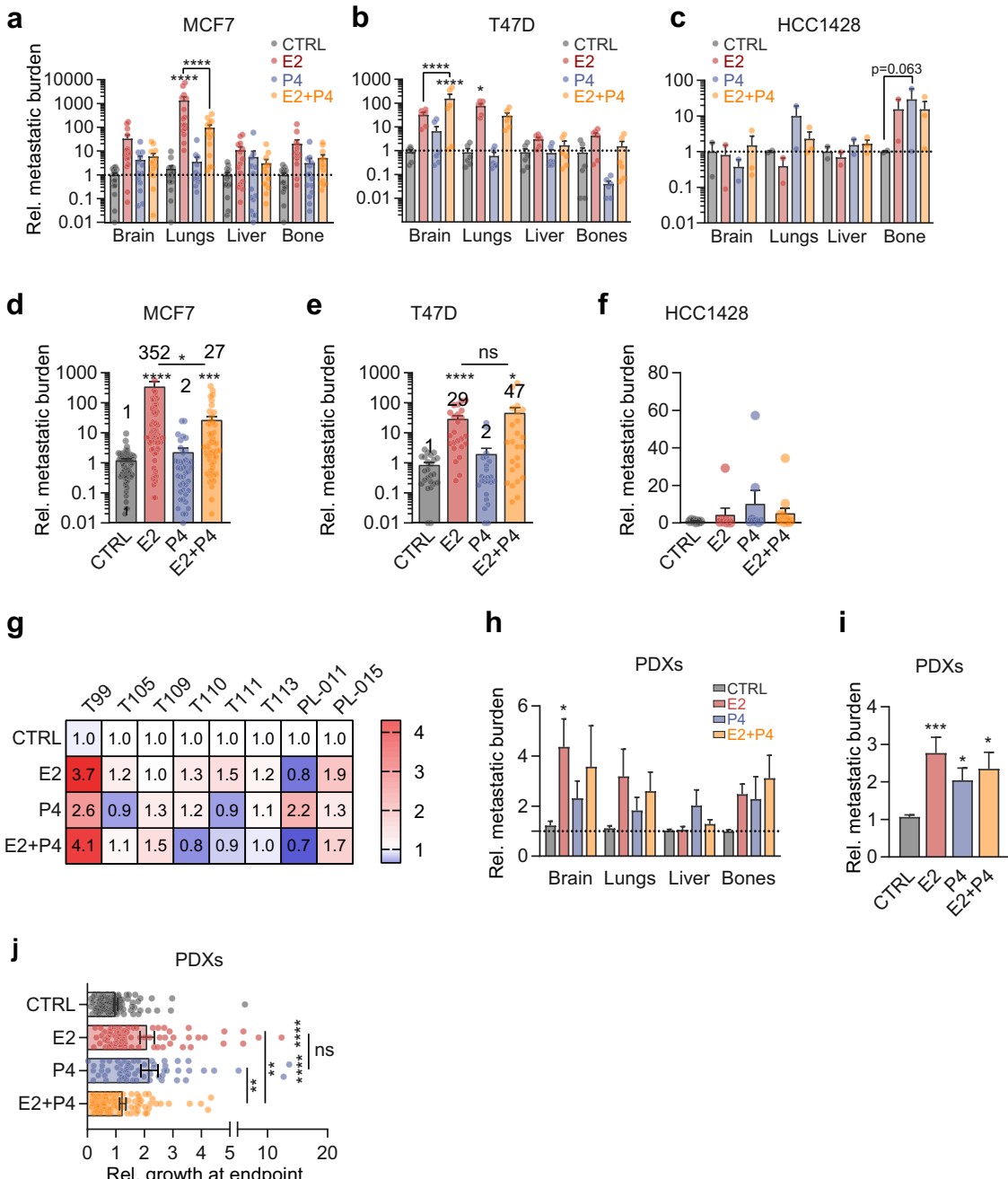

**Fig. 3 Hormone exposure increases metastatic spread. a–c** Relative ex vivo radiance in different organs of mice engrafted with **a** MCF7 median $n = 15$. **b** T47D median $n = 6$. **c** HCC1428 median $n = 2$ after 45-day-hormone treatment. Bar graphs indicate average ± SEM; two-way ANOVA with Dunnett's multiple comparison test. **d–f** Bar graphs showing relative metastatic burden in different organs of MCF7, T47D, and HCC1428 xenograft-bearing mice treated with E2, P4, or E2 + P4 average ± SEM. Kruskal–Wallis test, followed by Dunn's multiple comparison test. **d** CTRL $n = 59$, E2 $n = 60$, P4 $n = 44$, E2 + P4 $n = 56$. **e** CTRL $n = 24$, E2 $n = 24$, P4 $n = 24$, E2 + P4 $n = 24$. **f** CTRL $n = 8$, E2 $n = 8$, P4 $n = 8$, E2 + P4 $n = 12$. **g** Heatmap showing relative metastatic load in mice xenografted with different patients' tumors. Data represent mean fold change radiance compared to CTRL for each PDX. Friedman test, followed by Dunn's multiple comparison test. T99, CTRL $n = 3$, E2 $n = 6$, P4 $n = 4$, E2 + P4 $n = 6$. T105, CTRL $n = 5$, E2 $n = 4$, P4 $n = 4$, E2 + P4 $n = 5$. T109, CTRL $n = 3$, E2 $n = 6$, P4 $n = 4$, E2 + P4 $n = 4$. T110, CTRL $n = 4$, E2 $n = 2$, P4 $n = 4$, E2 + P4 $n = 3$. T111, CTRL $n = 3$, E2 $n = 3$, P4 $n = 3$, E2 + P4 $n = 2$. T113, CTRL $n = 3$, E2 $n = 2$, P4 $n = 2$, E2 + P4 $n = 1$. PL-011, CTRL $n = 3$, E2 $n = 2$, P4 $n = 3$, E2 + P4 $n = 2$. PL-015, CTRL $n = 4$, E2 $n = 4$, P4 $n = 4$, E2 + P4 $n = 3$. **h** Bar graph of relative metastatic burden per organ in mice engrafted via teats with different PDXs after 60-day-hormone treatment. Data represent mean ± SEM. CTRL $n = 24$, E2 $n = 21$, P4 $n = 22$, E2 + P4 $n = 21$, mixed effects analysis, and post hoc multiple comparisons. **i** Bar plot of relative metastatic burden in PDXs bearing mice under hormone treatment. Data represent mean ± SEM. CTRL $n = 96$, E2, P4, E2 + P4 $n = 84$. Holm–Sidak's multiple comparisons test. **j** Bar graph of relative growth at endpoint in mice engrafted with PDXs. Data represent mean ± SEM. CTRL $n = 24$, E2 $n = 21$, P4 $n = 22$, E2 + P4 $n = 21$, mixed effects analysis, and post hoc multiple comparisons.

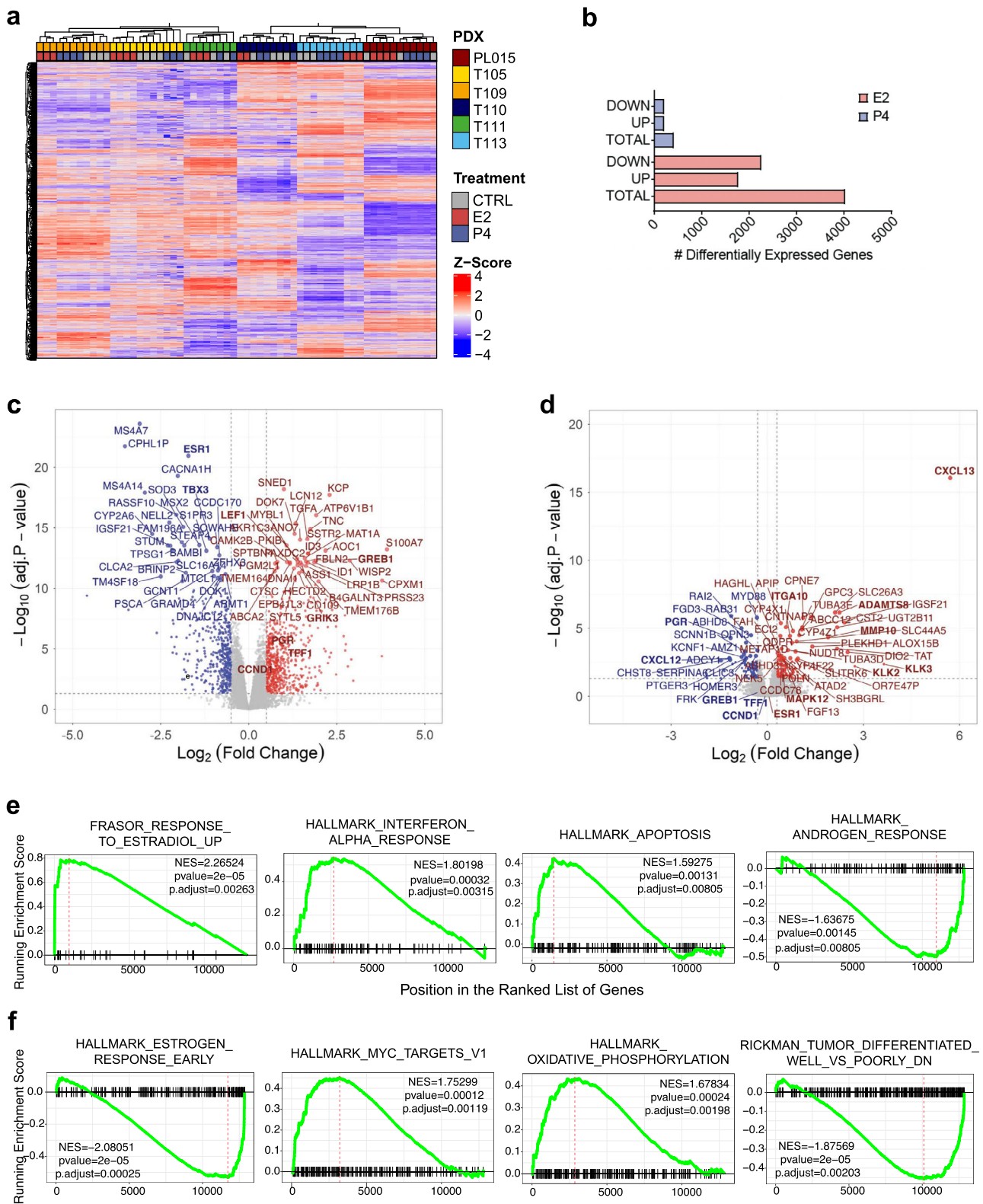

**Fig. 4 Transcriptional response of patient-derived tumor cells to E2 and to P4. a** Heatmap showing unsupervised hierarchical clustering of the 500 most variable genes across six PDX samples treated with E2 or P4 or untreated. Normalized expression levels were scaled to Z-scores for each gene, $n = 69$. **b** Bar plot showing the number of genes differentially expressed across six different PDXs in response to E2 or P4, adjusted $p$ value cutoff of 0.05, logFC < −0.5 or logFC >0.5. **c**, **d** Volcano plot showing differentially expressed genes in response to E2 (**c**) or P4 (**d**) across six different PDXs, each 3-4 biological replicates, $n = 69$. All highlighted genes have $p$ values <0.05 according to the limma model used for differential expression analysis. Genes with log2(FC) >0.5 in red and log2(FC) <0.5 in blue. Names of selected genes are indicated. **e**, **f** GSEA showing enrichment of pathways differentially regulated in response to E2 (**e**) or P4 (**f**). The red dashed line indicates NES normalized enrichment score.

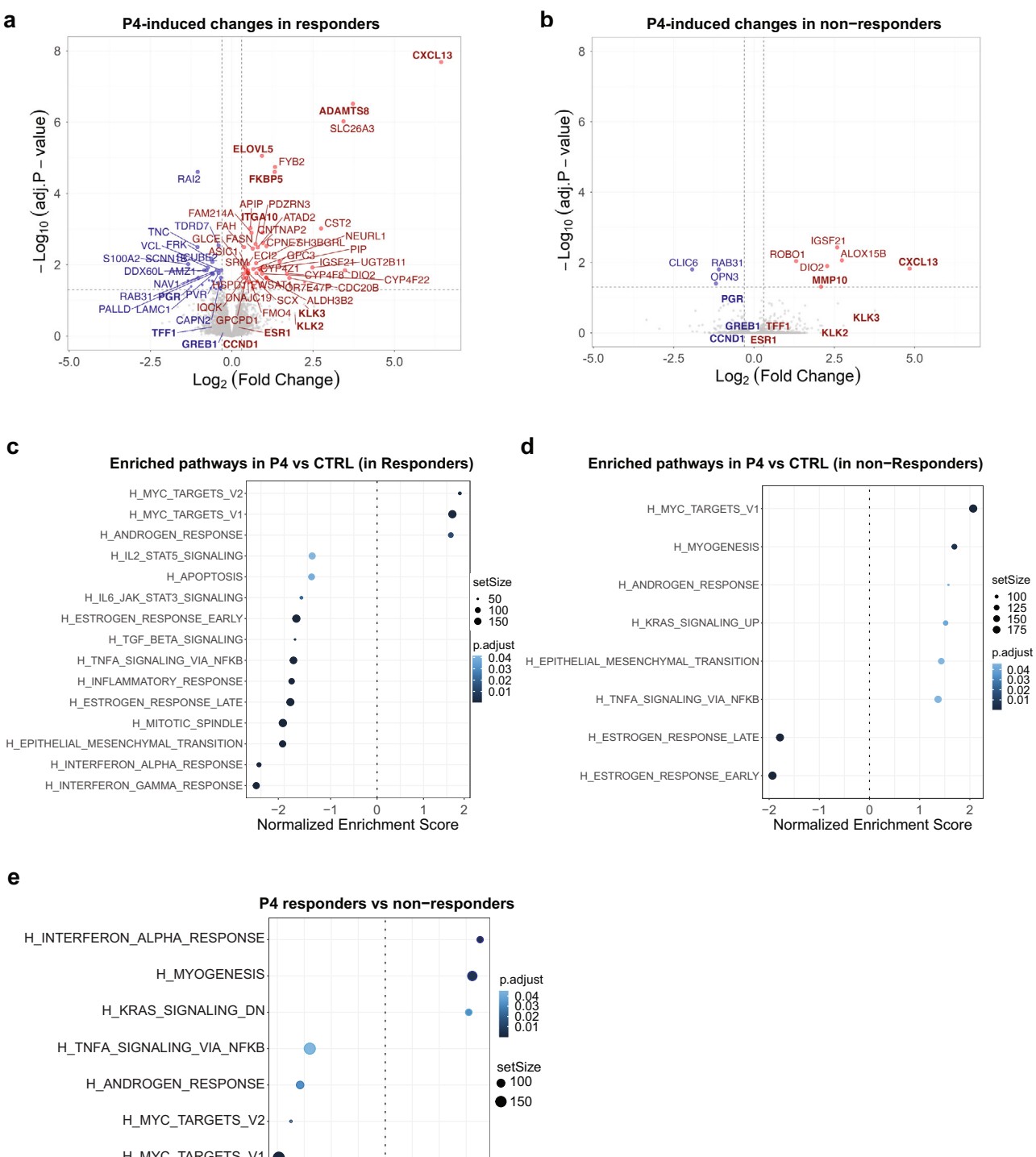

**Fig. 5 Transcriptional control of the response to P4. a, b** Volcano plots showing differentially expressed genes in response to P4 in responders **a** and non-responders **b**, three different PDXs, each 3-4 biological replicates. All highlighted genes have *p* values <0.05 according to the limma model used for differential expression analysis. Genes with log2(FC) >0.5 in red and log2(FC) <0.5 in blue. Names of selected genes are indicated. **c, d** Dot plots showing differentially enriched hallmark gene sets upon P4 stimulation in tumors that respond with proliferation to P4 (responders) (**c**) versus tumors that do not (non-responders) (**d**), three different PDXs, each 3-4 biological replicates. **e** Dot plot showing differentially enriched hallmark gene sets in unstimulated tumors that respond with proliferation to P4 (responders) versus tumors that do not (non-responders).

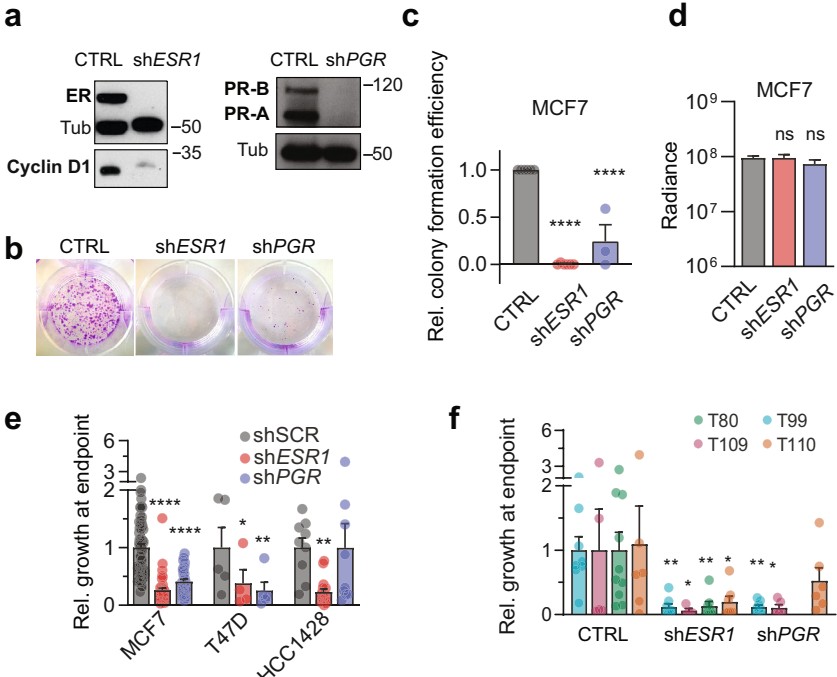

**Fig. 6 PR is necessary for tumor cell proliferation. a** ER and PR immunoblots of different MCF7 cell strains. **b** Representative pictures of colony formation assays of CTRL, sh*ESR1*, and sh*PGR* MCF7 cells 10 days after plating 1000 cells **c** Bar plot of respective quantification. Data represent mean ± SEM. CTRL = 6, sh *ESR1* = 6, sh *PGR* = 3. One-way ANOVA followed by Dunnett's multiple comparison test. **d** Radiance from glands injected with MCF7 cells, in which either *ESR1* or *PGR* were silenced, on day 1 after intervention. Data represent mean ± SEM. CTRL = 74, sh *ESR1* = 34, sh *PGR* = 24. Two-way ANOVA followed by Dunnett's multiple comparison test. **e** Relative tumor growth at endpoint of three ER + BC cell lines expressing sh*ESR1*, sh*PGR* or CTRL 90 days after injection. Data represent mean ± SEM. Two-way ANOVA followed by Dunnett's multiple comparison test. MCF7 CTRL *n* = 60, sh*ESR1 n* = 40, sh*PGR n* = 29; HCC1428 CTRL *n* = 5, sh*ESR1 n* = 4; sh*PGR n* = 5; T47D CTRL *n* = 9, sh*ESR1 n* = 17; sh*PGR n* = 8. **f** Bar plot showing relative tumor growth at endpoint of four PDXs expressing sh*ESR1* or sh*PGR* 140 days after injection. Data represent mean ± SEM. Two-way ANOVA followed by Dunnett's multiple comparison test. T99 *n* = 8 each; T109 CTRL *n* = 5, sh*ESR1 n* = 4; sh*PGR n* = 5; T80 CTRL *n* = 10, sh*ESR1 n* = 7; T110 CTRL *n* = 6, sh*ESR1 n* = 7, sh*PGR n* = 6.

resistance, we stably expressed *PGR* in MCF7 cells (MCF7:*PGR*) and concomitantly interfered with ER signaling. We did so either genetically using sh*ESR1* (Fig. 7a, b) or by mimicking clinical scenarios either pharmacologically by administration of fulvestrant, a selective ER degrader[38], or surgically by ovariectomy to simulate treatment with aromatase inhibitors. In vitro, ectopic *PGR* expression increased colony formation 1.5-fold under basal conditions but failed to rescue the loss of the colony-forming ability of sh*ESR1* cells (Fig. 7c). In vivo, MCF7:*PGR* cells grew 5-fold more than the control cells (Fig. 7d). While MCF7:sh*ESR1* cells showed decreased growth, sh*ESR1-PGR* cells grew like the control cells (Fig. 7d). At the endpoint, the Ki67 index was around 10% in the control tumors and increased to >60% in the MCF7:*PGR* even in the presence of coexpressed sh*ESR1* (Fig. 7e). IHC further showed that sh*ESR1* expression reduced the proportion of ER+ (Fig. 7f) and PR + cells (Fig. 7g) whereas the ectopic *PGR* expression increased the PR index from 20 to >90% (Fig. 7g).

Next, we treated mice engrafted with MCF7:control or MCF7:*PGR* cells with the widely used selective ER degrader, fulvestrant, for 70 days. MCF7:*PGR* injected glands developed palpable tumors. At the endpoint, mammary glands from untreated control mice showed invasive foci, whereas in fulvestrant-treated control mice most tumor growth was limited to the in situ stages (Fig. 8a, b). The MCF7:*PGR* cells invaded extensively, independent of fulvestrant treatment (Fig. 8a, b). IHC revealed an expected decrease in ER and PR expression by fulvestrant (Fig. 8b, c) and confirmed high PR expression in the MCF7:*PGR* cells (Fig. 8b, c). Fulvestrant decreased radiance at the endpoint of MCF7:control xenografts but not of MCF7:*PGR*

xenografts that grew twofold more than the MCF7:control xenografts (Fig. 8d). *PGR* overexpression increased metastatic load, both with and without fulvestrant treatment, 8 and 25-fold, respectively (Fig. 8e).

Next, mice injected with either MCF7:control or MCF7:*PGR* cells were ovariectomized to mimic the effect of clinical aromatase inhibition. In the absence of ovarian hormones, the growth of MCF7:control cells but not of the MCF7:*PGR* cells was decreased (Fig. 8f–h). IHC showed abrogation of PGR expression by ovariectomy (Fig. 8g, h). MCF7:*PGR* cells showed a trend for increased metastasis in the hormonally-ablated hosts (Fig. 8i) and increased metastasis to the brain (Fig. 8j). Thus, PR is sufficient to induce tumor cell proliferation, invasion, and metastasis when ER signaling is abrogated, making it an attractive potential target for therapeutic intervention, particularly in endocrine-resistant diseases.

## Discussion

Since G.T. Beatson discovered in 1895 that oophorectomy improved BC outcome[39], the role of ovaries in BC pathogenesis has attracted attention. Here, we provide new insights into the respective contributions of the two major ovarian hormones, E2 and P4, using highly disease-relevant models. We build on the recent demonstration that **m**ouse **in**tra**d**uctal (MIND) xenografting[26] enables the in vivo growth of ER + BC cells by providing the right microenvironment[29]. The tumor cells grow without the exogenous hormone supplementation required in the traditional subcutaneous fat grafts. As a result, the endocrine milieu in the host animals is clinically more relevant than in previous models because endogenous E2 levels compare to those

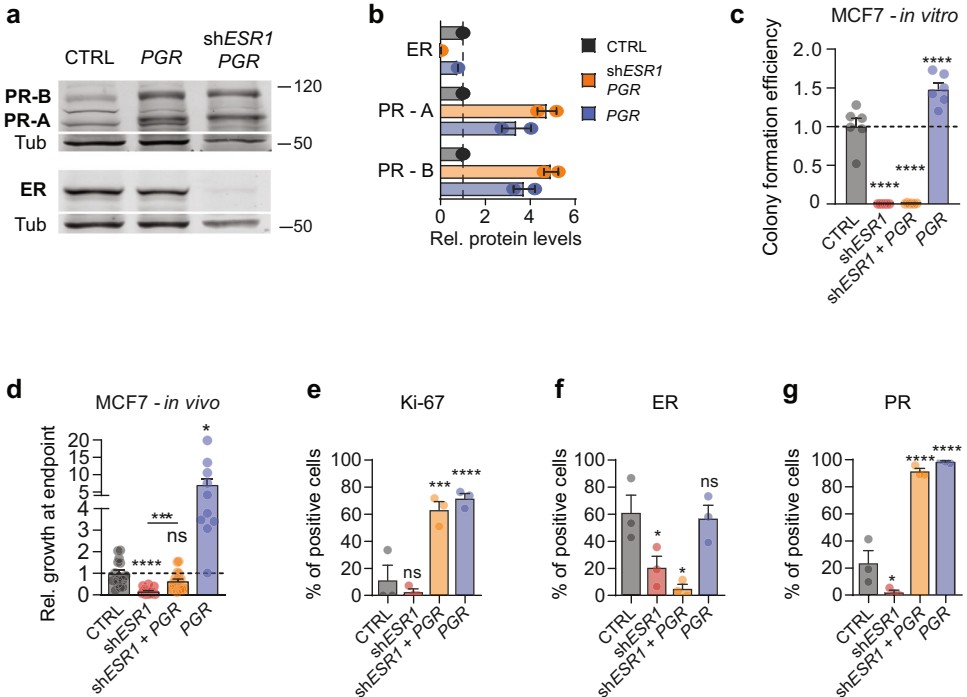

**Fig. 7 Ectopic PR expression overcomes ER abrogation. a** ER and PR immunoblot of CTRL MCF7 cells, or MCF7 cells upon *PGR* overexpression and *ESR1* silencing combined with *PGR* overexpression. **b** Quantification of ER, PR-A, and PR-B protein levels in MCF7 strains shown in **a**, n = 2 for all conditions. Data represent mean ± SEM. **c** Bar plots showing colony formation efficiency of sh*ESR1*, sh*ESR1* + *PGR*, and *PGR*-overexpressing MCF7 cells, n = 6 for all samples. Data represent mean ± SEM. **d** Relative in vivo growth at the endpoint of different MCF7 strains. Data represent mean ± SEM. CTRL n = 15, sh*ESR1* n = 18, sh*ESR1*-*PGR* n = 19, *PGR* n = 10. Brown–Forsythe and Welch ANOVA tests with Dunnett's multiple comparisons. **e–g** Quantification of protein levels measured by immunohistochemistry of Ki67, ER, and PR, respectively, in MCF7 strains grown in vivo intraductally. One-way ANOVA with Holm–Sidak's multiple comparisons test, n = 3 for all conditions. Data represent mean ± SEM.

found in postmenopausal women, in whom most BCs occur. We find that experimental exposures to E2 and P4, at levels observed during menstrual cycles, promote ER + BC progression. This is in line both with the clinical observations that BC is more aggressive in pre- than in postmenopausal patients and with epidemiological studies correlating the number of menstrual cycles women experience during their lifetime with their BC risk[2].

The altered immune function of the *NSG* host mice, which lack B- and T-cells and have reduced NK cell function is a major limitation of the model in light of the important role of the immune system in carcinogenesis and cancer therapy. However, we find that various inflammatory and immune pathways are modulated in the xenografts by E2 and P4 consistent with, in particular P4, having an immune suppressive role during pregnancy when it protects from the immune rejection of the fetus bearing paternal antigens. These observations highlight that innate immune functions in the *NSG* hosts persist and that they are hormonally controlled and may be functionally important. This is surprising because we selectively analyzed the human RNA sequence recovered from the xenografted glands. Although not formally excluded in the present study, it is unlikely that human immune cells contained in the patient-derived tumor cell graft persist over the 15 months after the initial injection. This suggests that the pathways are downmodulated in the human carcinoma cells as a result of paracrine interactions with mouse immune cells in the ductal microenvironment. Hence, tissue macrophages and dendritic cells from the *NSG* host may functionally interact with the human grafts and this crosstalk is under endocrine control. Furthermore, the absence of the T cell response may not be of major importance for preclinical studies with ER + xenografts because of this subtype has a low antigenic load, and in fact, clinical trials with checkpoint inhibitors had

negative outcomes[40,41]. The progress made in humanizing the mouse[42] provides the possibility to ultimately validate findings in the present models in immune intact hosts.

Hurdles to the widespread use of MIND ER + PDXs remain. There are logistic challenges to setting up and maintaining a clinical pipeline. Furthermore, the size of the tissue samples that can be obtained from ER + tumors without interfering with the diagnostic process is an issue. Pathologists can frequently spare less than 1 cm$^3$ and often <2 million cells are obtained following tissue dissociation. The proportions of carcinoma versus stromal and normal breast epithelial cells in the sample is uncertain. To obtain enough tumor cells for experimentation with multiple replicates, the initial grafts need to be amplified over 2 to 3 successive transplant generations, which can take up to 2 years because of long doubling times. As up to four glands can readily be engrafted, multiple data points on primary tumor growth can be collected per mouse. For the analysis of metastasis, only one data point per animal can be obtained. For this reason, we were unable to conclude on patient tumor-specific metastatic patterns in the present study.

Furthermore, growth rates differ from one patient sample to the next; hence experimental plans need to be individualized. For all these reasons, the faster-growing and more readily manipulatable cell line xenografts remain helpful in the initial testing of hypotheses. Moreover, the four cell lines included here were more sensitive to hormone stimulation than the patient-derived cells making it easier to discern biological responses. The increased amplitude of the hormone response may be a result of adaptation to culture and/or reflect a higher tumor cell homogeneity due to the selection of cell subpopulations when it is likely that in the early passage of patient-derived samples, multiple subclones exist, which have different responses to the hormones.

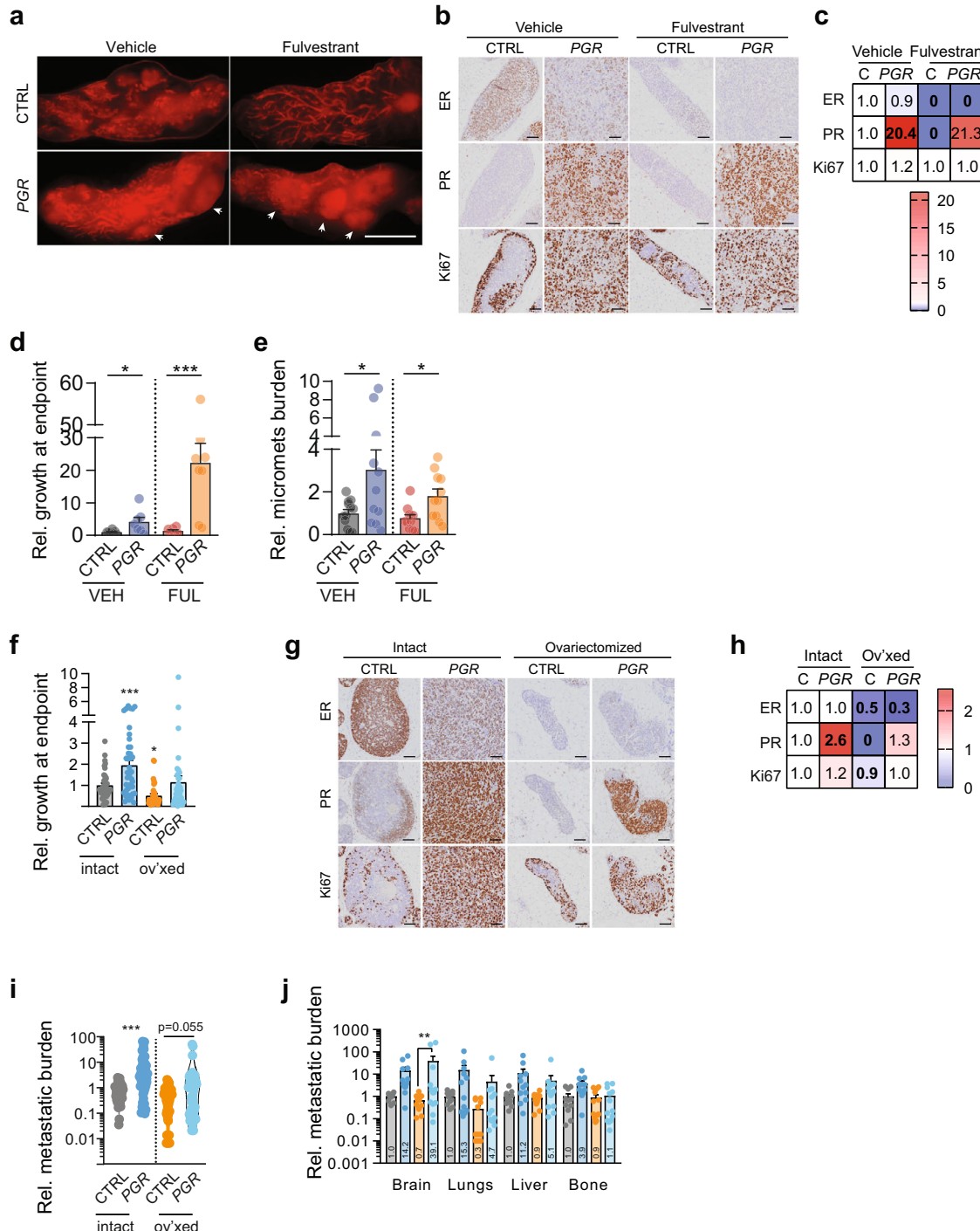

Interestingly, the most widely used MCF7 cells stand out from the other cell line models in that they grow faster and are exquisitely sensitive to E2. Their proliferative response to E2 is strikingly inhibited by co-administration of P4, whereas in T47D, HCC1428, and MDA-MB134-VI cells, the two hormones synergize in stimulating tumor cell proliferation. This urges caution in extrapolating findings in the most commonly used MCF7 cells to ER + BC in general.

While most ER + cell line models are derived from pleural effusions, i.e., late-stage disease the present study is largely based on patient-derived tumor cells mostly from untreated primary tumors. We find surprising heterogeneity in the sensitivity of individual patients' cells to E2 and P4 and the combination of the two hormones. In light of the long disease development and the heterogeneity of driver and oncogenic mutations, the genomic and epigenomic level and signaling context are likely patient-specific and result in patient-specific biological outcomes that may ultimately require tailored endocrine therapy.

Albeit to a different absolute and relative extent, in most patient samples, E2 elicits increased cell proliferation. In response to P4 just over half of the patient samples show a pronounced transcriptional response and proliferate. These patient-specific effects of P4 stimulation may account for some of the conflicting views about the role of the hormone in tumor progression. Two transcriptional signatures, which predict a lack of response to P4 in a PDX are high MYC activity and AR response both of which

**Fig. 8 Ectopic PR expression bestows resistance to endocrine therapy. a** Representative stereomicrograph images of mammary glands injected with MCF7 CTRL or *PGR*-overexpressing cells treated with vehicle or fulvestrant, scale bar: 1 cm. Arrowheads point to invasive areas. **b** Representative micrographs of IHC for ER, PR, and Ki67 MCF7 control and constitutive *PGR* expressing xenografts treated with either vehicle or fulvestrant. **c** Heatmap showing relative indices of marker positivity. Data represent fold change cell positivity compared to CTRL. $n = 3$ for all groups. Two-way ANOVA was followed by Tukey's multiple comparison test. For values printed in bold, $P < 0.05$. **d** Bar plots showing a relative change in radiance measured by bioluminescence imaging at the endpoint of MCF7 CTRL and MCF7 *PGR*-overexpressing cells in control and fulvestrant-treated mice. Two-way ANOVA with Tukey's multiple comparison test, median $n = 7$, average $n = 7$. [VEH CTRL $n = 9$, FULV CTRL $n = 7$, VEH PGR $n = 5$, FULV PGR $n = 7$. **e** Relative metastatic burden in mice engrafted with MCF7 CTRL and MCF7 *PGR*-overexpressing cells, treated with vehicle or fulvestrant. Mann–Whitney test, median $n = 11.5$, average $n = 11.5$. **f** Bar plots showing a relative change in radiance measured by bioluminescence imaging at the endpoint of MCF7 CTRL and MCF7 *PGR*-overexpressing cells in control and ovariectomized (ov'xed) mice, $n = 38, 43, 34,$ and 36. Data represent mean ± SEM. Two-way ANOVA with Tukey's multiple comparison test. **g** Representative micrographs of IHC for ER, PR, and Ki67 MCF7 control and constitutive *PGR* expressing xenografts in intact or ovariectomize host mice. **h** heatmap showing relative indices of marker positivity. Data represent fold change cell positivity compared to CTRL. $n = 5$ for all groups. Two-way ANOVA was followed by Tukey's multiple comparison test. For values printed in bold, $P < 0.05$. **i** Relative metastatic burden measured by bioluminescence imaging at the endpoint, $n = 44, 47, 47,$ and 50. Data represent mean ± SEM. **j** Relative metastatic spread to different organs was measured by bioluminescence imaging at the endpoint. Bars are colored according to the groups specified in **g, h**. Data represent mean ± SEM. Two-way ANOVA with Tukey's multiple comparison test. $n = 11, 43, 12,$ and 13 mice.

can be induced by P4 and may hence be a consequence of high PR signaling activity in these tumors that precludes further activation by progesterone. The effect on MYC may be direct[43] or indirect, mediated f.i. by increased Wnt signaling downstream of progesterone-induced Wnt signaling[44], which in turn induces MYC expression. In either case, the baseline PR signaling activity is independent of the level of the ligand, the levels of which fluctuate in all the host mice. This is in line with PR having ligand-independent scaffolding actions in signaling and transcriptional complexes with ER and PELP[15] and MAP kinase activation enabling PR engagement with accessible genomic sites[45]. Further experiments with more ER + PDXs will need to address whether MYC activity also determines the metastatic potential of the tumor cells and whether the observed differences in response to P4 stimulation also translate into different sensitivities to PR signaling abrogation.

Our finding that non-responders show increased TNFα signaling upon P4 exposure raises the intriguing possibility that macrophages, a major source of TNFα, are more active in the non-responders and may have a role in suppressing ER + tumor cell proliferation. The observation that a transcriptional signature of EMT, which is itself controlled by global histone3K36 methylation[46] is downmodulated in P4 responders and upregulated in non-responders argues that epigenetic factors may determine, at least this aspect of P4 response. As breast cancers develop over decades and acquire multiple divergent drivers and passenger mutations each tumor likely has a different signaling and epigenetic context both of which can affect the outcome of hormone receptor signaling and contribute to the patient-specific responses to E2 and P4.

The combined E2 and P4 treatment failed to increase PDX growth at the primary sites but resulted in an increased metastatic burden. This highlights a disconnect that has important implications for preclinical studies; these are mostly based on measuring tumor volume when it is ultimately the metastatic disease that makes cancer incurable and often lethal.

The metastasis-promoting effects of E2 and P4 treatment, the increased invasiveness of PR overexpressing tumors together with the genetic evidence that PR is required independently of ER for in vivo tumor growth of patient-derived cells argues that PR may be an important protein to target in ER + BC. The observation that ectopic PR expression is sufficient to drive tumor growth and even metastasis when ER signaling is genetically or pharmacologically abrogated in MCF7 cells points to the possibility that this may also apply in the context of endocrine resistance, in particular in tumors with *ESR1* mutations, which as a result of overactivated ER signaling show high PR expression levels. As most of

the clinically tested selective PR modulators lack specificity and show activity in particular versus the glucocorticoid and androgen receptor with substantial side effects, drugs targeting PR for degradation, such as PROTACs[47], may provide an attractive alternative. The intraductal xenografting approach offers new opportunities to test these hypotheses.

## Methods

**Cell culture**. MCF7 and T47D cells were cultured in DMEM 10% FBS, HCC1428 in RPMI 10% FBS, penicillin-streptomycin 1%. Cells were spin infected at 2250 rpm for 2.5 h and incubated with the lentivirus overnight. Cell lines were selected with 2 µg/ml puromycin for silencing experiments, while h*PGR* over-expressing cells were selected with blasticidin (2 µg/ml). Positive control of selection efficiency was always carried out in parallel on non-infected cells. Colony formation assays were performed in six-well-plates, seeding 1000 cells/well in a phenol red-containing medium. Ten days later, colonies were fixed with 4% PFA and stained with 0.5% crystal violet. For sg and sh plasmids see Supplementary Table 1.

**Clinical samples**. The Commission cantonale d'éthique de la recherche sur l'être humain approved the studies (45-05 and 72-04), and informed consent was obtained from all subjects. Tumor or tissue was obtained from the pathologist, mechanically and enzymatically dissociated[29], and dissociated cells were transduced with *ffLuc2/eGFP* lentivirus under control of the cytomegalovirus promoter at 25 °C for 2.5 h at 2500 rpm[48].

**Animals**. NOD.Cg-*Prkdc*^scid^ *Il2rg*^tm1Wjl^/SzJ mice were purchased from Charles River. All animal experiments were performed in accordance with protocols approved by the Service de la Consommation et des Affaires Vétérinaires of Canton de Vaud, Switzerland (VD 1865.3, 1865.4, and 1865.5). Mice were maintained and handled according to Swiss guidelines for animal safety with a 12-h-light-12-h-dark cycle, controlled temperature and food and water in polysulfone bottles ad libitum in SPF conditions, cages enriched with nesting material, cardboard, and wood tunnels, 12 h light cycle, 7 a.m. to 7 p.m. Animals are housed in IVC polysulfone cages—Green line—from Tecniplast®, type II long. Bedding is provided by Aspen Tapvei® (little squares about 4 mm × 4 mm × 1 mm). Water is acidified (pH between 2.5 to 3) on a resin column (Prominent® CH system). Diet from Provimi-Kliba® (cat# 3242, irradiated). Housing room temperature is 22 ± 2 °C, humidity 55 ± 10%.

**Xenograft procedures**. Eight to 12-week-old *NSG* females were anesthetized by intraperitoneal injection with 10 mg/kg xylazine and 90 mg/kg ketamine (Graeub) and injected into the cleaved teat with a blunt end Hamilton syringe (cat. no. HAMI80508), specifications: 50 µl 705 N, gauge 30/13 mm/pst3) with 100,000 cells for cell lines and 250,000 cells for patient-derived cells, and 400,000 for MDA-MB-134-VI cells. Dafalgan was administered intraperitoneally Temgesic® (Buprenorphinum) at 100 mg/kg when needed. Live imaging was performed from the day after injection with Xenogen IVIS Imaging System 200 (Caliper Life Sciences) upon intraperitoneal injection of 100 µl of luciferin (15 mg/ml) (Biosynth, cat# L-8220). Eight minutes after injection, mice were anesthetized with oxygen combined with 2% isoflurane, bioluminescence was measured from 12 min after injection. To examine metastatic spread, mice were injected with 300 µl luciferin (15 mg/ml) (Biosynth, cat# L-8220), tumors and organs of interest were dissected within 30 min, and imaged with IVIS (Perkin Elmer). Bioluminescence of treated mice

organs was normalized to the average bioluminescence of matched control organs. Slow-release pellets were fabricated by mixing silicone elastomer with a mixture of silicones, Part A (MP3745/E81949) and part B (MP3744/E81950) of low consistency silicone elastomer (MED-4011) and hormone powder[49,50] see Supplementary Table 2.

**IHC staining and markers quantifications**. Each xenografted mouse had one entire or at least half a gland harvested, fixed with PFA and paraffin-embedded. Sections were cut, dewaxed, rehydrated, and subsequently stained with antibodies using Ventana Discovery ULTRA (Roche Diagnostics, Rotkreuz, Switzerland) with Ventana solutions. Paraffin sections were pretreated with heat using standard condition (40 min) CC1 solution. The following primary antibodies were used with specific dilutions and incubation times: ER, Zytomed System, cat# BRB053, ready to use, incubation 16 min; PR, Ventana, cat# 790-2223, ready to use, incubation 32 min; Ki67 Abcam, cat# M3060, dilution 1:400, incubation 60 min; pHH3, Abcam, cat# ab5176, dilution 1:5000, incubation 16 min. After incubation with a rabbit Impress HRP antibody (Vector laboratories), the chromogenic revelation was performed with a ChromoMap DAB kit (Roche Diagnostics, Rotkreuz, Switzerland). Slides were scanned with an Olympus VS120-L100 slide scanner using a 20x/0.75 objective connected to a Pike F505 C Color camera. Images were loaded into QuPath[51] using the BioFormats Extension (https://github.com/qupath/qupath-bioformats-extension) and tumor cell regions were identified using a custom script. In short, RGB images were converted to Hue Saturation Brightness and the Brightness image was used to detect the tissue regions. A median filter of radius 3 pixels was applied, followed by ImageJ's Triangle auto-threshold method [https://imagej.net/Auto_Threshold#Triangle]. Only regions larger than 1e4 square microns were kept for analysis. The regions were then manually curated to ensure accurate detection of the tissue of interest. Individual cells in each region were segmented using QuPath's built-in Watershed Cell Detection algorithm (detection performed on: Optical density sum, Pixel Size": 0.2 µm, Background Radius": 8.0 µm, Median Radius: 0.0 µm, Sigma: 1.0 µm, minArea: 10.0 µm, maxArea: 400.0 µm, Threshold: 0.05, maxBackground: 2.0, Watershed Post Process: true, Cell Expansion: 0.0 µm, include Nuclei, Smooth Boundaries, Make Measurements, Threshold Compartment used was set on nucleus: DAB OD mean; Nuclear DAB OD mean >0.2 was considered positive). The number of cells positive for HR and proliferation markers expression was counted, and the average values were plotted.

**Western blot**. Western blots were performed using primary antibodies against ERα (Santa Cruz, sc-543), PR (Santa Cruz, sc-7208), and Cyclin D1 (Neomarkers, #RB-212-P0). The secondary antibody used against γ-tubulin (Sigma, clone GTU-88).

**LC-MS hormone measurements**. Sample preparation, steroid extraction, liquid chromatography-mass spectrometry, and data processing as in ref. [52] with the following difference: after LC-MS injection for P4 analysis, the leftover sample was dried down under N2 flow, and estrogens were derivatized by resuspending in 1 mg/ml dansyl chloride in 50% acetone- in Na carbonate pH 10.2 for 15 min at 65 °C before re-injection for estrogen analysis.

**Whole mounts**. Mammary gland whole mounts were performed as described previously[53]. Stereo micrographs were imaged by LEICA MZ FLIII stereomicroscope with a Leica MC170 HD camera, and fluorescence images were acquired with LEICA M205FA fluorescence stereomicroscope equipped with a Leica DFC 340FX camera.

**RNA sequencing and raw data processing**. MCF7 xenografts treated with hormone pellets were dissociated into single cells, mouse cells were depleted with immunomagnetic beads (Miltenyi, cat#130-104-694). For MCF7 xenografts with *PGR* overexpression, whole glands were sequenced, and mouse and human reads were separated computationally[54]. Raw reads were aligned to the human (hg38) and mouse genome (mm9) using HISAT2 (v2.1.0)[55], the exact parameters are hisat2 -k 5 -p 4—seed 42. Gene counts were generated using FeatureCounts[56] and data preprocessed with the edgeR package from Bioconductor[57]. The Voom function[58] of the limma package from Bioconductor[59] was used to normalize the data for sequencing depth differences, estimate the mean-variance relationship of the log counts, and generate a precision weight for each observation so that data were ready for the limma linear fitting function (lmFit).

**Statistical analysis**. Statistical analysis was performed with GraphPad Prism version 8.0.0 for Windows, GraphPad Software, San Diego, CA, USA, www.graphpad.com. Tests as indicated in figure legends. All statistical tests are two-tailed. Data were shown as means ± SEM, or as otherwise specified. Statistical significance is indicated as follows *$p < 0.05$, **$p < 0.01$, ***$p < 0.001$, ****$p < 0.0001$, • weak significance, n.s. not significant. When applicable, batch effects corresponding to different mammoplasties were modeled via the design matrix specifications (linear mixed effects modeling). Genes were considered differentially expressed based either on a $p$ value cutoff ($p < 0.05$) and fold change ($\log2(FC) | > 0.5$) or, when possible, an adjusted $p$ value cutoff of $p.adj < 0.05$. Pathway enrichment analysis was carried out using ClusterProfiler[60,61] from

Bioconductor (using default parameters). GSEA analysis was carried out using the GSEA function in ClusterProfiler, and the following annotated gene sets from MSigDB v6.2:[62,63] the Hallmark gene set[64], and the C2 curated gene sets.

**Reporting summary**. Further information on research design is available in the Nature Research Reporting Summary linked to this article.

## Data availability

The data generated in this study are provided in the Source Data file. The authors declare that all data supporting the findings of this study are available within the article from the authors upon reasonable request. The RNA-seq data generated and analyzed in this study have been deposited in the Gene Expression Omnibus (GEO) database under the accession codes: GSE192808, for PGR overexpressing MCF7 intraductal xenografts, GSE192809 for hormone-treated MCF7 intraductal xenografts, and GSE192810 for hormone-treated intraductal breast cancer PDXs.

## Code availability

The code employed for the analyses during the current study is open-source and available through the abovementioned packages in R. Scripts can be provided upon request.

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

## Acknowledgements

We thank P. Aouad, P. Dotto, M. Beato, and G. Vincent for critical reading of the manuscript, J. Dessimoz at the EPFL histology core facility, T. Laroche at the EPFL bioimaging and optics platform (BIOP), and B. Mangeat at the EPFL gene expression core facility (GECF) and C. Pulver for technical assistance. We extend our gratitude to the patients who participated in our study. V.S. and F.D.M. were supported by SNF "Exploring key steps of the metastatic cascade in ER+ breast cancer in vivo" 310030_179163/1 C.B., by Swiss Cancer League "Different facets of estrogen receptor alpha (ER) signaling during ER+ breast carcinogenesis" KFS-4738-02-2019-R C.B., A.A. by the ISREC foundation, and G.S. and G.A. by Biltema ISREC Foundation Cancera Stiftelsen, Mats Paulssons Stiftelse, and Stiftelsen Stefan Paulssons Cancerfond C.B. C.B. has support from H2020-MSCA-ITN (ITN-2019-859860-CANCERPREV).

## Author contributions

Conceptualization V.S. and C.B.; Formal analysis V.S., G.A., F.D.M., An.A., and C.L., Investigation V.S., Ay.A., L.B., F.D.M., C.S., and G.S.; Resources D.J., A.T., K.Z., A.S., M.F., Writing V.S. and C.B.; Funding acquisition P.B. and C.B.

## Competing interests

The authors declare no competing interests.
