## [Peer Review File · Nature Communications]

Estrogen receptor positive breast cancers have patient specific hormone sensitivities and rely on progesterone receptorEditorial Note: This manuscript has been previously reviewed at another journal that is not operating a transparent peer review scheme. This document only contains reviewer comments and rebuttal letters for versions considered at *Nature Communications*.

REVIEWERS' COMMENTS

Reviewer #1 (Remarks to the Author):

Scabia and coworkers have used the physiologically relevant MIND model to demonstrate the impact of estrogen and progesterone (alone or combined together) on the growth and importantly, metastatic properties and spread of breast cancer cell lines and patient derived tumor cells in immunocompromised mice. They demonstrate in multiple models that both hormones are capable of increasing tumor growth and when combined, aid metastatic spread.

The mechanistic studies are timely and important, and provide badly needed clarity to the controversial question of targeting multiple steroid hormone receptors (both ER and PR) to prevent dangerous ER+/PR+ breast cancer progression. Notably PR (in the absence of ER) was sufficient to induce tumor cell proliferation, invasion, and metastasis. These studies demonstrate a genetic role for PR in advanced deadly cancer behaviors and reveal a path to new biomarkers of response for agents that could be used to block PR.

Typo on line 174 (MYC targets)

Dr. Christy Hagan has recently published immuno-modulatory actions of PR in mice models and her work should be cited here.

PMID: 33958486

See also/cite how PR inhibits interferon via degradation of STAT2:

PMID: 32391191

Reviewer #4 (Remarks to the Author):

I was asked to comment on authors' responses to concerns from Reviewer 3.

In my opinion, the Reviewer 3 raised some excellent points, and the majority of them were addressed by the authors. Complete response to some concerns would have required additional experiments which the authors chose not to do. While this might not be satisfactory, it is understandable due to the complexity and length of the work required to perform these in vivo studies. It might be helpful to comment on some of these limitations and the plans to address them in future studies in the Discussion, especially the grouping of the samples based on metastatic spread (comment #9) and concurrent RNA seq and phenotypical studies in a set of shPR knockdown models (comment #10).

RESPONSE TO REVIEWER COMMENTS:

We thank **reviewer 1** for appreciating the physiological relevance of the models and the importance of our work.

We checked for the typo “MYC targets” but could not find it.

The two references were added on page 7 to the end of the first paragraph.

We thank **reviewer 4** for stepping in and appreciating the complexity and length of the work required to perform the in vivo studies.

We now discuss that the two points raised by reviewer 3 (9 and 10) will be address by future studies as enough samples are sufficiently expanded to address them, page 11 to the end of the second paragraph.